# Orthogonal navigation of multiple visible-light-driven artificial microswimmers

Jing Zheng[1], Baohu Dai[1], Jizhuang Wang[1], Ze Xiong [1], Ya Yang[1], Jun Liu[1], Xiaojun Zhan[1], Zhihan Wan[1] & Jinyao Tang [1]

Nano/microswimmers represent the persistent endeavors of generations of scientists towards the ultimate tiny machinery for device manufacturing, targeted drug delivery, and noninvasive surgery. In many of these envisioned applications, multiple microswimmers need to be controlled independently and work cooperatively to perform a complex task. However, this multiple channel actuation remains a challenge as the controlling signal, usually a magnetic or electric field, is applied globally over all microswimmers, which makes it difficult to decouple the responses of multiple microswimmers. Here, we demonstrate that a photoelectrochemically driven nanotree microswimmer can be easily coded with a distinct spectral response by loading it with dyes. By using different dyes, an individual microswimmer can be controlled and navigated independently of other microswimmers in a group. This development demonstrates the excellent flexibility of the light navigation method and paves the way for the development of more functional nanobots for applications that require high-level controllability.

[1] Department of Chemistry, The University of Hong Kong, Pokfulam, Hong Kong, China. Jing Zheng and Baohu Dai contributed equally to this work. Correspondence and requests for materials should be addressed to J.T. (email: jinyao@hku.hk)

The design and fabrication of microswimmer[1–4] with flexible manipulation capability and excellent biocompatibility may have a profound impact on people's lives due to its potential applications in healthcare[5,6], manufacturing[7–9] and environmental remediation[10–14]. Over the past decade, varieties of artificial microswimmers have been developed which generate propulsion thrust by harvesting energy from magnetic field[15–17], electric field[18,19], acoustic wave[20–23], light[24–28] or chemical fuels[29–33]. On the other hand, to control the migration of the microswimmers remotely, an external control signal is required to align the microswimmers to the desired orientation. Currently, most of the demonstrated microswimmers rely on the external magnetic field for direction control due to its simplicity and the excellent biological tissue penetration. However, one challenge facing the magnetically driven microswimmer is that since it is difficult to confine the magnetic field in a small area, a global field is applied over all microswimmers within the interested area, which makes it difficult to address the microswimmer individually and realize multi-channel actuation[16]. This multi-channel capability is particularly important for the long-envisioned non-invasive surgery as well as some micromanipulation application where many conceived operations can only be accomplished by the cooperative maneuvers of many individually addressed microswimmers. Recently, some limited success of the multi-channel actuation of magnetic microswimmers were proposed/demonstrated by introducing another degree of freedom to differentiate magnetic actuators such as the different mechanical resonance frequency[34,35], different magnetic moment[16] or spatially addressable substrate[36]. However, none of these attempts can completely decouple the response of different microswimmers in the arbitrary environment, which may limit the potential application of these microswimmers. Compared to the magnetic navigation, light navigation[26,28,37,38] is an emerging method to manipulate the microswimmer based on controlled thermophoresis or the shadowing effect. Particularly, if the microswimmer can be programmed with orthogonal spectral response[39], the multichannel actuation can be readily achieved as multiple beams of light with different wavelength can be utilized as independent control signals.

Recently, we designed the hierarchical Janus titania/silicon (TiO$_2$/Si) nanotree as artificial microswimmer with excellent controllability[26]. However, due to the large bandgap of titania, the microswimmer is only sensitive to UV radiation which is not preferred for biomedical application.

Inspired by the well-developed dye-sensitized solar cells (DSSCs) which can harvest energy from visible light[40–42], here, we demonstrate a general strategy to produce light-driven artificial microswimmers which can be propelled and navigated with visible light. After loading the titania with a photosensitizer (organic dyes), the visible photons can be absorbed by the sensitizer, and the photocurrent is generated by the electron injection from the photoexcited sensitizer to the titania nanowires. As a result, the spectral response of the nanotree is dominated by the absorption spectrum of the corresponding dyes. In this work, three commonly used dyes for dye-sensitized solar cells were employed to realize the nanotree based microswimmers with blue, green and red light peak sensitivity, respectively. We demonstrated that the photoresponse of the microswimmers could be completely decoupled and the two-channel actuation with minimum interference is realized. This strategy is a general method for multichannel actuation without the requirement for specialized substrate or minimum swimmer–swimmer distance. We expected that by applying different dyes with narrower absorption band and near-infrared activity, the microswimmers with more orthogonal channels and near-infrared activity could be realized. Furthermore, our strategy can also be applied to prepare microswimmers with broadband visible light sensitivity[43,44], which can be used as the solar-powered microswimmer for environmental remediation.

## Results

**Fabrication and characterizations of the microswimmers.** Large-scale Si/TiO$_2$ nanotree was synthesized by modified metal-

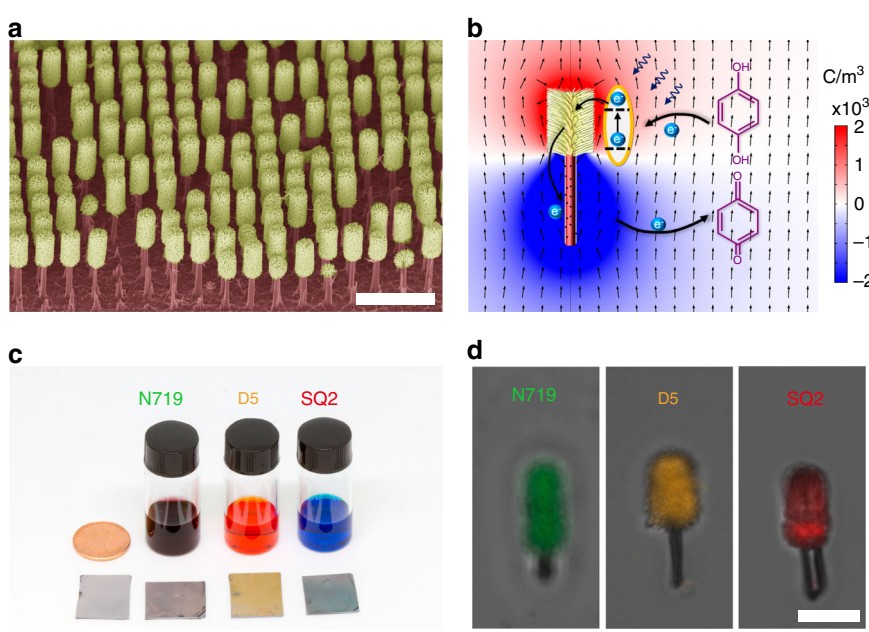

**Fig. 1** Schematic diagram and structural characterizations of the microswimmers. **a** False-colored scanning electron microscope image of the dye-sensitized Janus nanotree with TiO$_2$ nanowire branches and silicon nanowire trunk (scale bar: 10 µm). **b** Schematic diagram of the dye-sensitized microswimmer driven by photoelectrochemical reaction with the numerically simulated charge distribution (color map). The length of the arrow is normalized and does not represent the flow magnitude. **c** The photograph of the dye alcoholic solutions and the corresponding dye-sensitized nanotree samples. **d** The confocal fluorescence mapping images of the dye-sensitized microswimmers, which show that all three dyes are selectively loaded onto the TiO$_2$ nanowire surface (scale bar: 5 µm)

assisted electroless chemical etching process followed by TiO$_2$ nanowire hydrothermal growth based on our previous study[26,45]. The scanning electron microscopy (SEM) image shows that the as-prepared nanotrees are composed of 6-µm-long silicon tails and 4-µm-long TiO$_2$ nanowire heads with 2.5 µm diameter (Fig. 1a). To extend the photoresponsivity of the nanotree microswimmer to visible range, three sensitizing dyes N719 (cis-diisothiocyanato-bis(2,2′-bipyridyl-4,4′-dicarboxylato) ruthenium(II) bis(tetrabutylammonium), as received from Solaronix), D5 (3-(5-(4-(diphenylamino)styryl)thiophen-2-yl)-2-cyanoacrylic acid,as received from dyenamo, Inc.) and SQ2 (5-carboxy-2-[[3-[(2,3-dihydro-1,1-dimethyl-3-ethyl-1H⁻ benzo[e]indol-2-ylidene)methyl]-2-hydroxy-4-oxo-2-cyclobuten-1-ylidene]methyl]-3,3-dimethyl-1-octyl-3H⁻ indolium, as received from Solaronix) were employed to code the microswimmers with green, blue and red light sensitivity, respectively. In typical DSSCs, the iodide/triiodide (I⁻/I$_3$⁻) redox couple is utilized to shuttle the charge from anode to cathode. Recently, it is discovered that the hydroquinone/benzoquinone (QH$_2$/BQ) can also serve as an effective redox shuttle for high-efficiency DSSCs[46], which can also be applied to our microswimmer system. As shown in Fig. 1b, our nanotree is immersed in QH$_2$/BQ mixture aqueous solution and functions as a miniaturized DSSCs, where the dye-sensitized TiO$_2$ branches serve as the photoanode and the silicon nanowire trunk serves as the photocathode. Upon illumination, the dye molecules absorbed energy from the incident light and transfer the photoexcited electrons to TiO$_2$ nanowires and promote the oxidation of QH$_2$ into BQ which releases H⁺ and makes the local solution around the TiO$_2$ nanowires slightly positively charged. Meanwhile, the photoexcited electrons in silicon nanowire reduced the BQ back to QH$_2$ which releases OH⁻ and makes the local solution around the silicon nanowire slightly negatively charged. This unbalanced distribution of the charged H⁺ and OH⁻ ions is simulated with

commercial numerical software (COMSOL Multiphysics, See the Methods) and shown in Fig. 1b. The nanotree propels by this self-generated electrical field via electrophoresis mechanism as reported previously[26]. A nanotree migration speed (SQ2 sensitized microswimmer under 660 nm light illumination with an intensity of 328 mW·cm⁻²) of ∼8.7 µm·s⁻¹ can be calculated by the fluid speed simulation using the method developed by Solomentssehv et al.[47,48] (see the Methods). Particularly, due to the selective chemisorption of the carboxylate acid (−COOH) group on TiO$_2$ surface, the organic dye is covalently loaded on the nanotree (Fig. 1c) by simply immersing the synthesized nanotrees into the dye alcoholic solutions. The selective anchoring and distribution of dyes were confirmed by the confocal fluorescence images (Carl Zeiss LSM 710 NLO) of the dye-sensitized nanotree. As shown in Fig. 1d, all dyes are selectively anchored on the TiO$_2$ surface and will not affect the photoelectrochemical properties of silicon nanowire.

**Wavelength and intensity-dependent migration.** Since the dye-sensitized microswimmer is propelled by the light absorption of dye molecules, the spectral response of the microswimmer is determined by the spectral response of the corresponding dye-sensitized solar cell. The external quantum efficiency (EQE) of the dye-sensitized nanowire solar cell with corresponding dyes was measured based on previously reported procedure[49] (see the Methods and Supplementary Fig. 1) and compared with the spectral response of the microswimmer. As shown in Fig. 2a, the peak EQE for D5, N719 and SQ2 sensitized solar cell are ∼450, ∼510 and ∼610 nm respectively. To quantify the spectral response of the dye-sensitized microswimmers, the migration speeds of the microswimmers are quantified while a supercontinuum laser coupled with the variable linear filter was utilized as the wavelength tunable illumination source. For all measurement, the

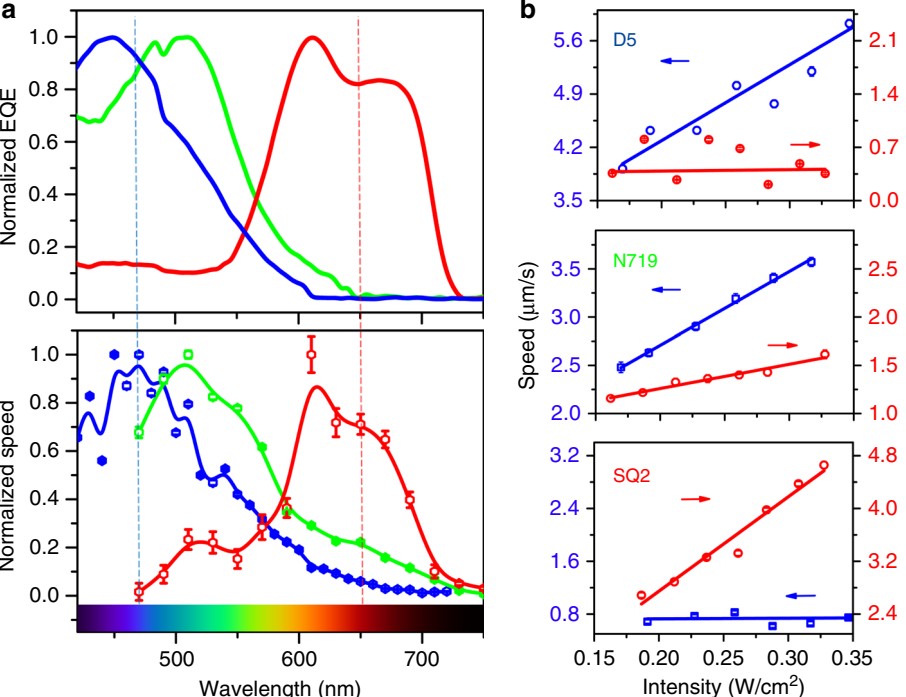

**Fig. 2** The spectral response of the dye-sensitized microswimmers. **a** The normalized EQE of the nanowire-based dye-sensitized solar cells (D5 (blue), N719 (green) and SQ2 (red)) compared with normalized spectral response of the corresponding dye-sensitized microswimmers. The vertical blue and red line indicated the illumination wavelength in **b**. **b** The absolute migration speed scales linearly with the illumination intensity of its corresponding absorbed light for various dye-sensitized microswimmers: D5, N719 and SQ2, respectively

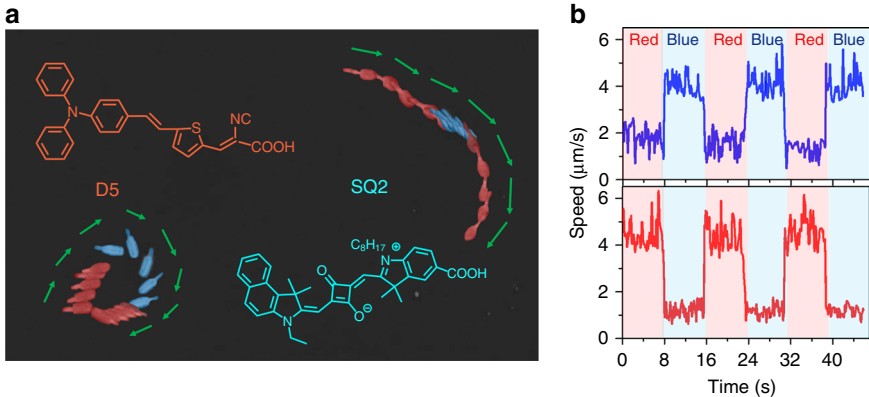

**Fig. 3** The orthogonal photoresponse of the dye-sensitized microswimmers. **a** The superimposed image of the sequential frames shows the migration of D5 and SQ2 sensitized microswimmers under blue (475 nm) and red (660 nm) light alternating illumination (Supplementary Movie 1). Inserts: the molecular structures of D5 and SQ2 dye. **b** The migration speed of D5 (blue curve) and SQ2 (red curve) sensitized microswimmers in **a** under the alternating light illumination

hydroquinone (100 mM)/benzoquinone (1 mM) mixture solution was selected as the reversible redox shuttle to replace the toxic $H_2O_2$ solution for better biocompatibility. As shown in Fig. 2a, the spectral response of the three dye-sensitized microswimmers matches well with the normalized EQE spectrum, confirmed the essential role of the dyes assisted PEC reaction in this microswimmer system. Particularly, the spectral response peak of D5/N719 and SQ2 sensitized microswimmer are well separated without significant overlapping which can support two independent channels for propulsion and navigation. Particularly, 475 nm (blue light) and 660 nm (red light) light-emitting diodes (LEDs) were utilized as the light sources (Supplementary Fig. 3) and the microswimmer migration speed is quantified at varying illumination intensities. To benchmark the intrinsic photoresponsivity of the microswimmer, the light-intensity-normalized migration velocity (LINMV), which is the slope of the absolute migration speed as a function of light intensity, is measured under 475 nm and 660 nm illumination. As shown in Fig. 2b, D5 dye can selectively absorb the blue light (475 nm) over the red light (660 nm), and the corresponding microswimmer show high LINMV $\left(v_{e(D5,475nm)} = 1.00 \pm 0.05 \, \text{mm}^3 \cdot \text{J}^{-1}\right)$ (95% confident interval) for blue illumination and low LINMV for red illumination $\left(v_{e(D5,660nm)} = 0.018 \pm 0.008 \, \text{mm}^3 \cdot \text{J}^{-1}\right)$ (95% confident interval). For SQ2 dye, due to the opposite absorption spectrum, a strong red light response $\left(v_{e(SQ2,660nm)} = 1.43 \pm 0.06 \, \text{mm}^3 \cdot \text{J}^{-1}\right)$ (95% confident interval) and a weak blue light response of the corresponding microswimmer is observed $\left(v_{e(SQ2,475nm)} = 0.01 \pm 0.03 \, \text{mm}^3 \cdot \text{J}^{-1}\right)$ (95% confident interval), which is suitable for orthogonal navigation when coupled with D5 sensitized microswimmer as shown in the next section. For N719 dye, which can absorb broader spectrum, both blue and red light response is observed $\left(v_{e(N719,475nm)} = 0.77 \pm 0.07 \, \text{mm}^3 \cdot \text{J}^{-1}, \right.$ $\left. v_{e(N719,660nm)} = 0.25 \pm 0.06 \, \text{mm}^3 \cdot \text{J}^{-1}\right)$ (95% confident interval), which promises a potential application of solar energy propelled microswimmers[44].

To further demonstrate the independent addressable autonomous migration of the dye-sensitized microswimmers, the mixture of D5 and SQ2 sensitized microswimmers are tested with alternating blue (475 nm) and red (660 nm) illumination. As shown in Fig. 3a and Supplementary Movie 1, upon illuminated with red light (660 nm), the D5 dye-sensitized microswimmer moves slowly due to the weak absorption (as shown in Fig. 2a), while the SQ2 dye-sensitized microswimmer shows a fast migration. When switched to blue illumination (475 nm), the D5 dye-sensitized microswimmer exhibits a high speed while the

migration of SQ2 dye-sensitized microswimmer is largely suppressed. The corresponding migration speed for the D5 and SQ2 sensitized microswimmers under alternating illumination is shown in Fig. 3b.

**Orthogonal navigation of the dye-sensitized microswimmers.** As we demonstrated previously[26], the nanotree-based microswimmer can not only be propelled but also be navigated by light with the self-shadowing effect of the nanotree. The multichannel controllability can be achieved with the light of different wavelength if the orthogonal spectral response of the microswimmers can be obtained. Here we demonstrate the independent navigation of multiple dye-sensitized microswimmers based on D5 and SQ2 dyes. This general strategy may also be extended to other organic dyes and inorganic quantum dots sensitized system to realize different spectral response of the microswimmer.

A customized stage is utilized to study the navigation ability of the D5 and SQ2 dye-sensitized microswimmers where four independently controllable red (660 nm) and blue (475 nm) LEDs are mounted at the four edges of the stage (see the Methods and Supplementary Fig. 2) to illuminate the microswimmers from controllable direction. As shown in Fig. 4a, the D5 and SQ2 dye-sensitized microswimmers are subjected to the side illuminations of 475 and 660 nm wavelength light spontaneously to control their orientation, while the 550 nm green illumination from the microscope objective is utilized for imaging which has minimum interference with both D5 and SQ2 sensitized microswimmers. Specifically, the 475 nm blue light (~45 mW·cm$^{-2}$) serves the control signal for D5 sensitized microswimmer and the interference for SQ2 loaded microswimmer, while the 660 nm red light (~70 mW·cm$^{-2}$) is the interference for D5 sensitized microswimmer and control signal for SQ2 sensitized microswimmer. In this two-channel control system, the signal-to-interference ratios (S/I, expressed in decibels) for D5 and SQ2 loaded microswimmers are −1.92 and 1.92 dB, respectively (see the Methods). To realize independent operation, the S/I should be greater than the interference tolerance which can be estimated as the ratio between LINMVs of the corresponding interference and the signal. Thus, the interference tolerance of the D5 and SQ2 loaded microswimmers can be calculated as −15.63 and −15.35 dB (see the Methods), respectively, which is well below the S/I applied in our system. As shown in Fig. 4b and Supplementary Movie 2, the 660 nm red illumination from fixed orientation is constantly lit up, while the 475 nm blue illuminations is sequentially lit up from various directions to rotate the

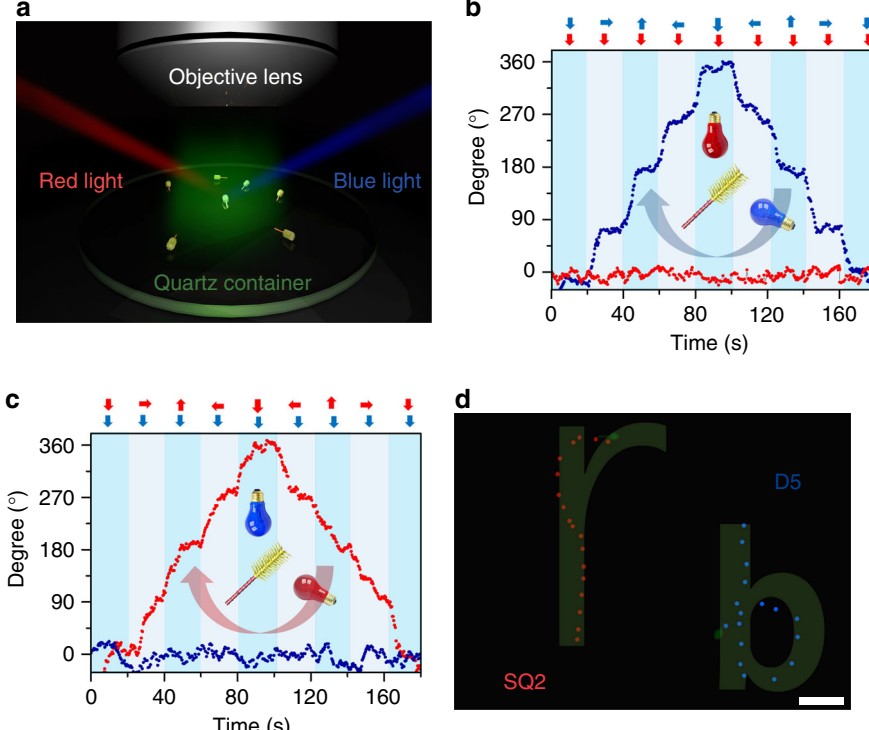

**Fig. 4** The orthogonal orientation control and navigation of the dye-sensitized microswimmers. **a** The schematics of the experimental setup where the blue (475 nm) and the red (660 nm) side illumination are used to control the orientation of the microswimmers while the green (550 nm) top illumination is used for imaging. **b** The angle of the D5 (blue) and SQ2 (red) sensitized microswimmers under fixed 660 nm illumination with rotating 475 nm illumination **c** The angle of the D5 (blue) and SQ2 (*red*) sensitized microswimmers under fixed 475 nm illumination with rotating 660 nm illumination. The red and blue arrows in **b**, **c** indicate the light propagation direction of the corresponding light. **d**. The trajectory of SQ2 (red) and D5 (blue) sensitized microswimmers spell 'r' and 'b' spontaneously as navigated with the blue and red light (scale bar: 20 μm)

D5 sensitized microswimmer irrespective of the red illumination direction. During this process, the orientation of SQ2 loaded microswimmer is locked by the illumination direction of 660 nm red light and does not show any rotation with the blue light source. Comparatively, the fixed 475 nm blue illumination with rotating 660 nm red illumination only drive the rotation of the SQ2 sensitized microswimmer without affecting the alignment of the D5 sensitized microswimmer (Fig. 4c and Supplementary Movie 2). These experiments demonstrated that the dye-sensitized microswimmer could indeed be controlled independently. As shown in Fig. 4d and Supplementary Movie 3, the D5 sensitized microswimmer (blue) and the SQ2 sensitized microswimmer (red) are simultaneously propelled and steered with 475 and 660 nm illumination and spell 'r' and 'b', respectively. With this simple dye-sensitize strategy, in principle, the microswimmer can be coded with an arbitrary spectral response by controlling the absorption spectrum of the adsorbed dyes. With the fruitful knowledge of the well-developed DSSCs and readily developed varieties of dyes and quantum dots, further improved functional microswimmers are expected. For example, more orthogonal channels for the microswimmers can be developed based on tandem dye-sensitized solar cell[50]. Furthermore, the near-infrared sensitive microswimmers can be developed on infrared dyes[51] which are preferred for biological application due to the larger optical penetration depth in tissues.

## Discussion

In summary, we have successfully designed and demonstrated the visible light driven dye-sensitized artificial microswimmers based on Janus TiO₂/Si nanotree structure. The spectral response of the microswimmer is determined by the absorption spectrum of the

loaded dye on the TiO₂ nanowires. By properly selecting the dyes with orthogonal absorption spectrum and the corresponding illumination wavelength, the sensitized microswimmers can be independently propelled and navigated. This orthogonal multi-channel navigation capability of the microswimmers represents a major step towards the more controllable microswimmers. More advanced nanorobot may be developed by integrating multiple independent controllable parts for complexed functions.

## Methods

**Fabrication procedure of dye-sensitized nanotree forest**. The large-scale nanotree forest was synthesized by modified metal-assisted electroless chemical etching process followed by TiO₂ nanowire hydrothermal growth. Prior to the dye adsorption, the prepared Janus TiO₂/Si nanotree forest was sintered in air under 450 °C for 30 min, and the platinum nanoparticles were loaded on the surface of the silicon trunk by dipping the prepared sample into the mixture solution of 0.5 mM chloroplatinic acid (Sigma-Aldrich) and 0.5 M hydrofluoric acid (Sigma-Aldrich) for 1.5 min for 6 cycles.

Three sensitizing dyes N719 (cis-diisothiocyanato-bis(2,2′-bipyridyl-4,4′-dicarboxylato) ruthenium(II) bis(tetrabutylammonium), as received from Solaronix), D5 (3-(5-(4-(diphenylamino)styryl)thiophen-2-yl)-2-cyanoacrylic acid, as received from dyenamo, Inc.) and SQ2 (5-carboxy-2-[[3-[(2,3-dihydro-1,1-dimethyl-3-ethyl-1H⁻ benzo[e]indol-2-ylidene)methyl]-2-hydroxy-4-oxo-2-cyclobuten-1-ylidene]methyl]-3,3-dimethyl-1-octyl-3H-indolium, as received from Solaronix) were employed to sensitize the as prepared nanotree forests. In a typical staining process, the samples were immersed in a 0.5 mM ethanolic solution of dye for 5 h to complete the dye adsorption. After rinsed with ethanol and dried under nitrogen flow, the samples were scraped off from the silicon substrate by a razor blade and transferred into redox couple solution for motion test.

**Charge distribution and electric field simulation**. Ion-induced charge distribution and fluid speed were simulated using the commercial COMSOL Multiphysics package. 2D models were adapted from the bimetallic motor system. Hydroquinone/benzoquinone (QH₂/BQ) system was taken as a representative system for numerical study. Cations (H⁺) and anions (OH⁻) are generated at TiO₂ (anode)

and Si (cathode) surfaces, respectively and are further distributed by the diffusion (Supplementary Table 1), convection and migration of ions (Equation Eq. (1)):

$$\nabla \mathbf{J}_i = \mathbf{u}\nabla c_i - D_i\nabla^2 c_i - \frac{z_i F D_i \nabla(C_i \nabla \phi)}{RT} \tag{1}$$

where $\mathbf{J}_i$ is the flux of ion $i$, $\mathbf{u}$ is the fluid velocity, $F$ is the Faraday constant, $\phi$ is the electrostatic potential, $R$ is the gas constant, $T$ is the temperature and $c_i$, $D_i$, $z_i$ are the concentration, diffusion coefficient, and charge of species $i$, respectively. The $H^+$ and $OH^-$ will react into $H_2O$ quickly when they combine with each other.

The electric field $\mathbf{E}$ ($\mathbf{E} = -\nabla\phi$) in Eq. (1) is calculated using the Poisson equation:

$$-\varepsilon_0 \varepsilon_r \nabla^2 \phi = \rho_e = F(Z_+ c_+ + Z_- c_-) \tag{2}$$

where $\varepsilon_0$ is the vacuum permittivity and $\varepsilon_r$ is the relative permittivity of water, $Z_+ = +1$ and $Z_- = -1$, $\rho_e$ is the volumetric charge density, $F$ is the Faraday constant and $C_+$ and $C_-$ are the concentration of the $H^+$ and $OH^-$, respectively.

The boundary condition was adopted according to the zeta potential of $TiO_2$ (~ +10 mV) and Si nanowire (Si surface ~ −20 mV). The ion fluxes generated on the anode and cathode surfaces are calculated based on SQ2 sensitized microswimmer illuminated by 660 nm light at 328 mW·cm$^{-2}$ illumination power and the EQE measured from the DSSCs (Supplementary Fig. 1), which is corresponding to the photocurrent density of $TiO_2$ (184 A·m$^{-2}$)(Eq. (3)).

$$J = \frac{I e \lambda}{h c}\eta \tag{3}$$

where $J$ is the photocurrent density, $I$ (328 mW·cm$^{-2}$) is the light intensity, $e$ is elementary charge, $\lambda$ (660 nm) is wavelength of incident light, $h$ is Planck constant, $c$ is the velocity of light and $\eta$ (10.5%) is the EQE of SQ2 measured at 660 nm. The simulated migration speed of ~8.7 µm·s$^{-1}$ is roughly agreed with the experimental result (~4.7 µm·s$^{-1}$, Fig. 2b)

**Dye-sensitized solar cell fabrication and characterization.** The external quantum efficiency (EQE) of the dye-sensitized nanowire solar cells were measured based on a two electrode system. Firstly, a fluorine-doped tin oxide (FTO) glass substrate covered with 2 µm long $TiO_2$ nanowire array was prepared under similar hydrothermal process as the Janus nanotrees. After sintering in air under 450 °C for 30 min, the $TiO_2$ coated FTO substrate was immersed in a 0.5 mM ethanolic solution of D5, N719 and SQ2 respectively for 5 h at room temperature to complete the dye adsorption (Supplementary Fig. 1a). These as-prepared dye-sensitized samples were utilized as photoanode. A Pt counter electrode utilized as photocathode was prepared by sputtering 5 nm Pt on another piece of FTO substrate with the same size of the photoanode. In addition, a mixture aqueous solution with 100 mM hydroquinone, 1 mM benzoquinone, and 0.1 M KCl was used as the electrolyte.

The external quantum efficiency (EQE) spectra were obtained as a function of wavelength from 420–750 nm using a computer-controlled system (EQE-R measurement system, Enli Technology Co. Ltd.) and the incident photon flux was determined with a calibrated silicon photodiode. As shown in Supplementary Fig. 1b, the peak EQE for D5, N719 and SQ2 sensitized solar cells are ~450, ~510, and ~610 nm, while the efficiency reach ~10, ~6 and ~12%, respectively.

**Microswimmer migration test procedure.** The solution with microswimmers was added into a customized liquid sample container made of two glass slides and observed using Olympus MX51 optical microscope. All videos were recorded by a digital video camera (Canon EOS 60D) at 1920 × 1080 resolution @ 30 fps.

The supercontinuum laser (SC-Pro, Wuhan Yangtze Soton Laser Co., Ltd.) coupled with the variable linear filter was used as the light source for migration speed measurement (Fig. 2a) from 420 to 760 nm with 20 nm bandwidth.

A customized stage (Supplementary Fig. 2a) containing a 3 W 475 nm LED bead (CREE XPE, blue light) and a 3 W 660 nm LED bead (red light) was used as the light source for intensity-dependent speed measurement (Fig. 2b) and multi-channel manipulation (Fig. 3 and Supplementary Movie 1).

Four 3 W 475 nm LED beads (blue light) and 3 W 660 nm LED beads (red light) were attached to a customized hollow aluminum box (Supplementary Fig. 2b). Two joysticks were utilized to control the illumination of the LEDs independently for the self-alignment test (Fig. 4b, c and Supplementary Movie 2) and independent navigation (Fig. 4d and Supplementary Movie 3).

**Signal to interference ratio analysis.** The intensity of LED light was measured using a standard Photodiodes (SM1PD1A, THORLABS). The intensity illuminated from the side blue LED and red LED (Supplementary Fig. 2b) are ~45 mW·cm$^{-2}$ and ~70 mW·cm$^{-2}$, respectively. Thus, the signal-to-interference ratios (S/I, expressed in decibels) for D5 and SQ2 loaded microswimmers are −1.92 dB and

+1.92 dB, respectively (Eq. (4)).

$$\begin{aligned} S/I_{(D5)} &= 10\log_{10}\left(\frac{\text{Intensity(blue)}}{\text{Intensity(red)}}\right) = -1.92\,\text{dB} \\ S/I_{(SQ2)} &= 10\log_{10}\left(\frac{\text{Intensity(red)}}{\text{Intensity(blue)}}\right) = 1.92\,\text{dB} \end{aligned} \tag{4}$$

The light-intensity-normalized migration velocity (LINMV) (95% confident interval) under blue and red light illumination are as follow:

$v_{e(D5,475nm)} = 1.00 \pm 0.05\,\text{mm}^3 \cdot J^{-1}$, $v_{e(D5,660nm)} = 0.018 \pm 0.008\,\text{mm}^3 \cdot J^{-1}$, $v_{e(SQ2,475nm)} = 0.01 \pm 0.03\,\text{mm}^3 \cdot J^{-1}$, $v_{e(SQ2,660nm)} = 1.43 \pm 0.06\,\text{mm}^3 \cdot J^{-1}$,

Thus, the minimum S/I (interference tolerance) for D5 and SQ2 sensitized microswimmers are -15.63 dB and -15.35 dB, respectively (Eq. (5)).

$$\begin{aligned} \text{Interference tolerance}_{(D5)} &= 10\log_{10}\left(\frac{V_{e(D5,660nm)}}{V_{e(D5,475nm)}}\right) = -15.63\,\text{dB} \\ \text{Interference tolerance}_{(SQ2)} &= 10\log_{10}\left(\frac{V_{e(SQ2,475nm)}}{V_{e(SQ2,660nm)}}\right) = -15.35\,\text{dB} \end{aligned} \tag{5}$$

**Data availability**. The data that support the findings of this study are available from the corresponding author upon reasonable request.

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

## Acknowledgements

This work was supported in part by the Hong Kong Research Grants Council (RGC) General Research Fund (GRF17305917, GRF17303015, ECS27300814), the University Grant Council (Contract No. AoE/ P-04/08), the URC Strategic Research Theme on Clean Energy (University of Hong Kong), and the URC Strategic Research Theme on New Materials (University of Hong Kong).

## Author contributions

J.Z. and B.D. contributed equally to this work. J.Z., B.D. and J.T. conceived and designed the experiments. J.Z., B.D., X.Z., J.L. and Z.W. fabricated the devices. J.Z. and B.D. performed the measurements and analysis the data. Z.X., J.W. performed the numerical simulation. J.Z. and Y.Y. took the confocal fluorescence mapping images, J.Z. and J.T. co-wrote the paper. All authors discussed the results and commented on the manuscript.

## Additional information

**Competing interests:** The authors declare no competing financial interests.

