## [Peer Review File · Nature Communications]

Reviewers' comments:

Reviewer #1 (Remarks to the Author):

This manuscript by Zheng and Dai et al. reports a new light-driven micromotor system that can be selectively activated by visible light of different frequency. This was achieved by modifying the micromotor surface with light absorbing dyes. As a result, micromotors of different dye modification can be activated independently by different light, and orthogonal control was possible. In addition, by taking advantage of the shadowing effect, micromotors can also be steered by light of different frequencies, although the quality of such steering still needs improvement.

In general, this is a high quality paper with novel and inspiring results. In particular, the knowledge from dye sensitized solar cell was implemented in the field of nano- and micromachines, and this is quite innovative and refreshing. Light absorption was therefore shifted from UV to visible spectrum, which offers great promises for certain applications. Data analysis was proper, and experimental details are adequate for readers familiar with the related techniques to reproduce these results. I therefore support its publication on Nature Communication, provided that the following minor issues are addressed properly.

1. Since not every reader of Nat Comm. understands the principles of dye sensitized solar cell, or electron transfer in semiconductors in general, can the authors explain in a little more detail how the dyes help the micromotors with visible light propulsion, while the original version can only absorb UV? In a typical DSSC system, electrons go through a dye-TiO₂-I⁻/I₃⁻ cycle. What is the equivalent of this cycle in the current system? And is there any net consumption of chemicals in the cycle, such as dyes or hydroquinone/benzoquinone, that limit the lifetime of these micromotors?
2. What is the zeta potential of the nanotree? Would it change during or after the operation? Does the absorption of dyes change the zeta potential? On a related note, can the authors comment on the directionality of these nanotrees, and compare it to self-electrophoresis theories?
3. The COMSOL simulation seems to be a little weak, since it does not provide any more useful information on the operation and steering of these nanomotors, nor does it help explain the roles of dyes. To make it more useful, can the authors calculate the motor speed based on the measured/estimated parameters (how much is the flux?), and compare it to experimental values? It is even better if the actual electron transfer process can be simulated, rather than an assumed flux of H⁺ and OH⁻.
4. The selection of scales in Fig 2b (right Y axis) is a little arbitrary and can be misleading. For example, the data for SQ2 seems to suggest a weak absorption of blue light vs red. Yet the absolute speed of micromotors in blue is faster than red below 0.3 W/cm². The authors are suggested to check these confusing data.
5. Would these dye-modified motor move in H₂O₂, or other redox couples?

6. The purpose of the long discussion of LINMV is not quite clear. Fig. 2a already does an excellent job at conveying the point that different motors operate at different light frequencies.
7. What is the significance of “signal to interference” ratio? Is it similar to signal to noise ratio? Is a low ratio better or worse? Is 0.64 and 1.55 good or bad?
8. “Practical applications” are mentioned a few times. Although I’m equally excited about the prospects these micromotors offer, the mention of biomedical applications needs more support. For example, visible light does not penetrate deep into tissues or through skins, and as a result infrared light is commonly used for medical purposes. The authors acknowledged this by mentioning that “near infrared sensitive microswimmers can be developed on infrared dyes”. Are there any environmental applications that the authors can envision?
9. Minor language issues: at the end of introduction: “opens up a new dimension to study the highly scientific important collective behavior of the active particle”.

Reviewer #2 (Remarks to the Author):

A clearly written paper that takes the work of "nano-tree" swimmers and looks to incorporate dye molecules to generate wavelength specific enhance motion effects.

By choice of dye molecule, and illumination, it is possible to activate the swimming mechanism of one class of swimmers specifically.

The paper is well written, and the data supports the authors claims.

I have two very small comments to make:

I would like to see the emission spectra for the LEDs overlapped with the absorption spectra.

Error: Line 318 "lit" - should this be "light" ?

I believe this to be worth publication in naturecomm.

Reviewer #3 (Remarks to the Author):

The manuscript reported by Tang and co-workers present the light control of multiple nanobots independently. These microswimmers are nanofabricated by electroless chemical etching and hydrothermal growth of TiO₂ and acquire a “nanotree” architecture. The work is well presented and the conclusions confirm by all the data collected throughout the manuscript.

The external control of artificial microswimmers has been a need over the recent years. As the authors correctly mention in the introduction, magnetic control can guide “all” microswimmers confined in the area where magnetic field is applied without discrimination. Thus, local and individual guidance is a challenge. Light control has been used recently for control of artificial microswimmers, however the multiple control of microswimmers at the same time in different directions is, to my knowledge, not reported so far. The novelty of this work is clearly justified although nanotrees-based swimmers and Hydroquinone/Benzoquinone fuel has been reported by the same group recently. This work may motivate other groups in the field of micro-nanomotors to implement this light control systems for a variety of microstructures.

Here, they modified microswimmers with dyes that enable the motion in wavelength dependent way, namely red and blue light independently. This is the main novelty of this work.

This reviewer suggests that this work is suitable for publication after addressing the following comments:

Although the fabrication method allows many nanostructures to be generated, the data presented here only shows a couple of nanotrees. IT would be important to depict statistical analysis on the accurate response, and the success yield of nanotress on light control.

Is it possible to control 2 nanotrees with red and 2 with blue light? Or it is just 1 and 1 system? In their previous work in Nature Nanotech. The authors showed swarming phenomena which could be controlled externally. Is it possible to separate swarms of different nanotrees and collect them in different locations

Reviewer #1 (Remarks to the Author):

Q1. Since not every reader of Nat Comm. understands the principles of dye sensitized solar cell, or electron transfer in semiconductors in general, can the authors explain in a little more detail how the dyes help the micromotors with visible light propulsion, while the original version can only absorb UV? In a typical DSSC system, electrons go through a dye-TiO₂-I⁻/I₃⁻ cycle. What is the equivalent of this cycle in the current system? And is there any net consumption of chemicals in the cycle, such as dyes or hydroquinone/benzoquinone, that limit the lifetime of these micromotors?

We appreciate for the suggestion and agree that the nanomotor community may not familiar with the concept of DSSCs. A brief introduction have been added on line54-60 with an additional citation about DSSCs is added as reference 42 .

It is true that most DSSC is using I⁻/I₃⁻ redox shuttle, however, the BQ/QH₂ can also be used as efficient redox shuttle. The BQ/QH₂ in DSSC has been reported previously (Angew. 51, 9896-9899 (2012)). A statement discussion has been added to line 96-99 to address this, and the reference is added. Since the BQ/QH₂ also performs as an efficient shuttle, no net chemical reaction is expected. Thus, the lifetime of our motor is not affected by BQ/QH₂.

Q2. What is the zeta potential of the nanotree? Would it change during or after the operation? Does the absorption of dyes change the zeta potential? On a related note, can the authors comment on the directionality of these nanotrees, and compare it to self-electrophoresis theories?

It is quite difficult to measure the overall zeta potential of nanotree due to the instability of the suspension. For most of the nanotrees, the overall zeta potential are weekly positive as the migration towards the silicon end of observed, while some overall negative nanotrees are occasionally observed as well which shows the migration towards the TiO₂ end as shown in the supplementary videos. As described in our previous paper (Nat Nano 11, 1087-1092 (2016)), the pristine nanotree is overall positively charged due to the positively charged TiO₂ head. The positive zeta potential of TiO₂ is originated from the basic nature of TiO₂ surface, where the H⁺ from the solution is preferentially absorbed. After dyes' absorption, the surface oxygen is partially occupied by the chemisorbed the dye molecules (particularly, the carbonic acid group), which limited the absorption of the proton, thus the overall zeta potential is lowered. Most of the nanotree is still overall positively charged, while the overall negatively charged nanotree is also observed. To completely control the zeta potential of the nanotree, another chemical modification is needed (Eg. AEEA treatment as shown in the previous paper, Nat Nano 11, 1087-1092 (2016)). However, since the zeta potential control is not the main point of this paper, the detailed study in this direction is not included in this study.

Q3. The COMSOL simulation seems to be a little weak, since it does not provide any more useful information on the operation and steering of these nanomotors, nor does it help explain the roles of dyes. To make it more useful, can the authors calculate the motor speed

based on the measured/estimated parameters (how much is the flux?), and compare it to experimental values? It is even better if the actual electron transfer process can be simulated, rather than an assumed flux of H⁺ and OH⁻.

We appreciate for this suggestion and agree that the motor speed simulation may provide some additional information to this study. We modified our simulation using 2D model, the fluid speed is simulated and added to the paper on line 109-112 and compared to the experimental data. The figure 1b have been updated. A motor speed of $\sim 8.69 \mu\text{m/s}$ is calculated based on an adapted method reported previously (*J. Am. Chem. Soc.* **135**, 10557-10565 (2013)). We estimated the total reaction flux by using the EQE measurement value from our DSSC measurement, we believe this would be a better way to estimate the flux than other simulation based method. A good agreement is obtained between the experiment and simulation. However, we would like to point out that according to the fluorescent imaging result, the dye loading amount does vary depending on the fabrication process and the sample storage. Since the flux may vary from sample to sample, the equivalent surface zeta potential can also vary based on different coating condition, the simulation can only give qualitative estimation of the actual propulsion.

Q4. The selection of scales in Fig 2b (right Y axis) is a little arbitrary and can be misleading. For example, the data for SQ2 seems to suggest a weak absorption of blue light vs red. Yet the absolute speed of micromotors in blue is faster than red below 0.3 W/cm^2 . The authors are suggested to check these confusing data.

We appreciated that the referee has carefully examined our data. And after check original data, we found that the SQ2 data in Figure 2b is mislabeled, so the red response is much greater than the blue response. We corrected the data in this new plot which shows higher speed for SQ2 in red light than in blue light. In this new plot, we also colored the Y-axis for clarity.

Q5. Would these dye-modified motor move in H_2O_2 , or other redox couples?

The motor cannot move in H_2O_2 because the H_2O_2 would oxidize the dyes quickly. Upon immerse the dye loaded motor in the H_2O_2 , the reaction started immediately, and the color of dyes goes away. We did find many other redox couples could also work in our system; we are drafting another report on how to select redox couples based on the PEC reactions and corresponding electrochemistry.

Q6. The purpose of the long discussion of LINMV is not quite clear. Fig. 2a already does an excellent job at conveying the point that different motors operate at different light frequencies.

We define the LINMV as the slope of the absolute migration speed as a function of light intensity. We believe it is a better value to quantify the intrinsic ability of the nanomotor to convert the photon energy into propulsion force as it normalized the light intensity as well as

canceled out the influence of the Brownian Motion. Moreover, this value is used to evaluate the inter-channel crosstalk in our system. Please refer to the response of Q7 for detail.

Q7. What is the significance of “signal to interference” ratio? Is it similar to signal to noise ratio? Is a low ratio better or worse? Is 0.64 and 1.55 good or bad?

In any multi-channel communication system, the inter-channel crosstalk should be suppressed as much as possible to ensure high-quality communication. In our system, two channels are included (red and blue light). It is important to benchmark how well our nanomotor can distinguish the signal from interference. We can estimate the interference tolerance as the ratio of the signal response and interference response of the microswimmer which are both quantified by the LINMV.

To realizing independent operation, the S/I should be much greater than the interference tolerance for both red and blue channels. The higher the S/I, the better operation and less interference are expected for the microswimmer.

In this paper, we can estimate the minimum S/I (interference tolerance) for D5 and SQ2 loaded microswimmers are -15.63 dB and -15.35 dB, respectively. The S/I selected in our experiment is -1.92 dB and 1.92 dB, which is within the tolerance limit. A detailed discussion is added to the line 213-219 in the main text and the supplementary information.

Q8. “Practical applications” are mentioned a few times. Although I’m equally excited about the prospects these micromotors offer, the mention of biomedical applications needs more support. For example, visible light does not penetrate deep into tissues or through skins, and as a result infrared light is commonly used for medical purposes. The authors acknowledged this by mentioning that “near infrared sensitive microswimmers can be developed on infrared dyes”. Are there any environmental applications that the authors can envision?

We agree that it is still very challenging to apply the microswimmers in biomedical applications as the working environment is much more complicated. Thus, we have weakened the biomedical application in our manuscript. Here, we mentioned the “practical application” because we believe the optical technique does offer some advantages for nanomotor, for example the multi-channel controllability as demonstrated in this paper. As for the infrared sensitivity, there are many readily available infrared dyes has been applied to DSSCs, we believe they can be directly applied to our system for infrared sensitive microswimmer which is more relevant to biological application in deep tissue. The reference 51 is one example for such dyes. Another example is MK245 (Dyes and Pigments,122, 2015, 272-279), which is only sensitive to infrared light.

We agree that there are many other applications such as environmental remediation, which can be applied if solar radiation can be used for nanomotor propulsion, although the multi-channel controllability offered in this paper is not particularly useful in this aspect. However, the strategy can be applied to make solar powered microswimmer which is suitable for environmental applications. Some comments are added to the manuscript(line 67-69).

Q9. Minor language issues: at the end of introduction: “opens up a new dimension to study the highly scientific important collective behavior of the active particle”.

We think this statement does not add too much value to the field, so we deleted this comment.

Reviewer #2 (Remarks to the Author):

Q1, I would like to see the emission spectra for the LEDs overlapped with the absorption spectra.

The emission spectra of LEDs have been added to Supplementary Information (Figure S3).

Figure S3. The normalized absorbance of ethanolic solution of dye (D5 (blue), N719 (green) and SQ2 (red)). The dotted curve indicated the normalized emission spectra of LED (475 nm (blue), 660 nm (red)).

Q2, Error: Line 318 "lit" - should this be "light" ?

We have replaced the “lit” with “lighted”.

Reviewer #3 (Remarks to the Author):

Q1, Although the fabrication method allows many nanostructures to be generated, the data presented here only shows a couple of nanotrees. It would be important to depict statistical analysis on the accurate response, and the success yield of nanotrees on light control.

We appreciate for this comment and agree that the precise control of our nanotree quality is important for practical application. In order to achieve the same response over large quantities of nanotrees, the nanotree morphology (the length of silicon, the coverage of TiO₂ head) as well as the dye loading amount needs to be well controlled. We did observe and measure many motors' migration which matched well with our claims. However, the detailed examination also reveals that our process does not provide the uniform morphology control of the nanotrees, particularly, the TiO₂ nanowire density as well as TiO₂ coverage which affect the dye loading amount significantly. Since the propulsion force in our system is generated from the light absorption from loaded dyes, we did find the migration speed, and the rotational speed varies among different nanotrees. Further work is in progress to improve the morphology control of the nanotrees as well as to identify the quantitative correlation between the nanotree morphology to the migration behavior. However, these data and the detailed discussion are out of the scope of this paper, and it thus excluded in this manuscript.

Q2, Is it possible to control 2 nanotrees with red and 2 with blue light? Or it is just 1 and 1 system? In their previous work in Nature Nanotech. The authors showed swarming phenomena which could be controlled externally. Is it possible to separate swarms of different nanotrees and collect them in different locations

As mentioned previously, due to the morphology difference, although we can control a group of microswimmers, the response time can be different across different swimmers. We are working on improving the fabrication process to achieve better uniformity in our system for collective behavior study.

The swarming phenomena can be demonstrated in this system, and it is indeed very interesting to explore the orthogonal swarming in the mixture solution. This work is in progress and is much more complicated to explain as many inter-swimmer interactions does exist in our system. We are studying the collective behavior systematically in the following project.

REVIEWERS' COMMENTS:

Reviewer #1 (Remarks to the Author):

The reviewer is in general satisfied with the authors' response. No further revision is needed. Since the rebuttal letter contains important information for understanding this work, the authors are recommended to publish the reviewer comments as well as response.

Reviewer #2 (Remarks to the Author):

I am happy with these corrections and modification.

I believe this body of work is worthy of publication.

Reviewer #3 (Remarks to the Author):

The authors added more information related to the questions from the Reviewers 1 and 2. However, most of the requests from Reviewer 3 as the authors indicate that there are planned experiments to improve the controllability and reproducibility of the nanotrees. The paper will be improved with the responses provided in the response letter, so this referee has no further requests.